# Synergy-Space Recurrent Neural Network for Transferable Forearm Motion Prediction from Residual Limb Motion

**DOI:** 10.3390/s23094188

**Published:** 2023-04-22

**Authors:** Muhammad Hannan Ahmed, Jiazheng Chai, Shingo Shimoda, Mitsuhiro Hayashibe

**Affiliations:** 1Department of Robotics, Graduate School of Engineering, Tohoku University, Sendai 980-8577, Japan; chai.jiazheng.q1@dc.tohoku.ac.jp; 2Graduate School of Medicine, Nagoya University, Nagoya 464-0813, Japan; sshimoda@ieee.org

**Keywords:** transfer learning, forearm motion prediction, kinematic synergies, prosthesis control, rehabilitation robotics

## Abstract

Transhumeral amputees experience considerable difficulties with controlling a multifunctional prosthesis (powered hand, wrist, and elbow) due to the lack of available muscles to provide electromyographic (EMG) signals. The residual limb motion strategy has become a popular alternative for transhumeral prosthesis control. It provides an intuitive way to estimate the motion of the prosthesis based on the residual shoulder motion, especially for target reaching tasks. Conventionally, a predictive model, typically an artificial neural network (ANN), is directly trained and relied upon to map the shoulder–elbow kinematics using the data from able-bodied subjects without extracting any prior synergistic information. However, it is essential to explicitly identify effective synergies and make them transferable across amputee users for higher accuracy and robustness. To overcome this limitation of the conventional ANN learning approach, this study explicitly combines the kinematic synergies with a recurrent neural network (RNN) to propose a synergy-space neural network for estimating forearm motions (i.e., elbow joint flexion–extension and pronation–supination angles) based on residual shoulder motions. We tested 36 training strategies for each of the 14 subjects, comparing the proposed synergy-space and conventional neural network learning approaches, and we statistically evaluated the results using Pearson’s correlation method and the analysis of variance (ANOVA) test. The offline cross-subject analysis indicates that the synergy-space neural network exhibits superior robustness to inter-individual variability, demonstrating the potential of this approach as a transferable and generalized control strategy for transhumeral prosthesis control.

## 1. Introduction

Amputation of the upper limb at any level can considerably affect an individual’s ability to perform the activities of daily living (ADLs). The proficiency for such activities decreases with higher amputation levels [1]. The general requirements of prosthetic users can be primarily summarized as intuitive control, ease of use, and sensory feedback [2,3]. With the advancements in robotics and sensor technologies, very sophisticated and state-of-the-art upper extremity prostheses such as the DEKA arm [4] and modular prosthetic limb [5] are currently available. However, one of the persistent drawbacks is the interface between the prosthetic device and the user, attributable to the growing gap between the control methods and hardware improvements for prosthesis development.

In the myoelectric control domain, which is the most widely used control approach for prosthetic arms [6], the number of input signals a user can provide is always less than the degree of control (DOC). The DOC refers to the number of functions of a prosthetic device controllable by the user. Hence, this issue is even more critical for the case of transhumeral amputees as they can only provide electromyography (EMG) signals from the upper arm. However, they must control a prosthesis with numerous active degrees of freedom (DOFs), such as a powered elbow, wrist, and hand. As feasible solutions, surgical innovations such as targeted muscle reinnervation (TMR) [6] and advanced signal processing techniques, such as pattern recognition [7], have been employed to classify more significant numbers of distinct commands from residual muscle activities.

Although such measures allowed transhumeral amputees better control over their multifunctional prosthetic arms, the control was slow, sequential, and unnatural as physiologically appropriate muscles were unavailable. Such counter-intuitive control strategies and a lack of functionality are often the reasons for the high rejection rates of these devices [8,9]. Furthermore, the calibration requirements and signal sensitivity issues associated with EMG-based control also challenge its continuous daily use.

Numerous investigations have been conducted to explore alternative solutions, such as those based on myokinetic signals [10] using the residual kinetic activity of the limb, ultrasound signals [11], mechanomyography [12] using vibrations caused by muscle contractions as the control signals, and the residual limb motion strategy [13,14]. Bio-inspired learning approaches based on studying the central nervous system (CNS) and human motor control abilities have also been employed to develop more natural and intuitive arm control strategies [15].

Concerning the target reaching task or control of the prosthetic elbow joint motion in transhumeral amputees, the ideal case would involve the prosthetic device acting as a natural extension of the human body. A promising scheme to achieve this is automatically controlling the prosthetic elbow joint based on the natural relationships between the arm joints. Analyses of such joint coordination approaches have shown evidence that recurrent patterns exist in the joint kinematics for upper limb movements while performing reaching or grasping tasks. For example, between the elbow flexion and humeral inclination during reaching [16], between the hand azimuth and movement direction during grasping [17], as well as a variety of many other arm movements in ADLs [18]. These patterns are referred to as synergies.

In the present study, we have also focused on a synergistic method for intuitive control of the prosthetic elbow joint for transhumeral amputees. The majority of previous such motion-based approaches using residual limb motion rely on ANNs [19,20] to identify and model the shoulder–elbow kinematic relationship, as inductive learning (IL) applied in [21] and radial basis function networks (RBFNs) used in [22,23]. The ANN is typically trained to map the shoulder kinematics (provided as input) to the elbow or forearm kinematics (provided as output) through supervised learning. To the best of our knowledge, in such previous approaches, no prior synergistic information is extracted from the motion data used for training the ANNs. This type of approach, in simplistic terms, can be represented as in Figure 1a and is hereafter referred to as “direct estimation”. However, the direct estimation method has a limitation due to its sensitivity to inter-individual variability. As a result, the performance of the model drops significantly when applied across multiple users, making it less suitable for generalization and transferability.

### Transfer Learning Framework for Transhumeral Amputees

This paper presents the concept of the synergy-space neural network, whereby we explicitly combine the kinematic synergies with the learning system to address the limitation of the direct estimation method, aiming for a more robust and transferable control strategy. Synergies have been observed to be repeatable and shared across similar tasks and subjects. Therefore, we extract the synergistic information from the movement data and incorporate only the most significant synergy components in the learning process, enabling more precise and efficient training of the ANN while taking advantage of the shared nature of these synergies enhancing the transferability of the model. The idea is to train the ANN to predict the corresponding activation signals (see Equation (Equation 1)), which estimates the forearm motion when combined with the extracted synergy matrix. Because the ANN is trained using the most significant synergy components, the synergy-space neural network can learn particular features common to the arm movement tasks, allowing for better cross-subject transferability. In addition, using a smaller number of synergy components aids the ANNs in learning good policies by reducing the dimensionality of the state space. Lastly, being a synergistic approach, it allows for the kinetically natural and energy-efficient motion estimation of the arm movements.

Figure 1b illustrates the overall workflow of the proposed synergy-space neural network approach. We first extract the spatial synergy components and their corresponding time-varying activation signals from the source data (see Equation (Equation 1)) using the principal component analysis (PCA). Long short-term memory (LSTM), a particular type of recurrent neural network (RNN), is then trained to predict the extracted time-varying activation signals based on input shoulder kinematics. Finally, the forearm motions are estimated using Equation (Equation 2), where the extracted activation signals *C* are replaced with the LSTM-predicted signals $C_{p}$. Introduced by Hochreiter and Schmidhuber [24], the reason for using LSTM is its ability to learn long-term dependencies. LSTMs have internal mechanisms called gates that regulate the flow of information to handle the vanishing gradient problem in RNNs, making them very suitable for time-series prediction such as motion data or, in our case, time-varying activation signals. This paper proposes and evaluates the synergy-space neural network for forearm motion estimation, comparing its performance with the direct estimation approach. The contributions made in this paper are as follows:The implementation of the proposed synergy-space neural network as a transferable model for forearm motion estimation using residual shoulder kinematics during horizontal reaching movements.Personalized LSTM Models Evaluation: To validate the proposed methodology and its better learning capability through a detailed comparison between the performance of the synergy-space neural network and the direct estimation approach.Cross-Subject Evaluation: To demonstrate the strength of the synergy-space neural network as a transferable decoder, indicating its ability to handle inter-individual variabilities.

The remainder of this paper consists of the following sections. Section 2 gives the background into synergies and their characteristics. Section 3 explains the materials and methods employed to achieve our objectives, describing the experimental protocol, synergy extraction, LSTM training and analysis, and evaluation strategy. Section 4 demonstrates the validity of the proposed method through the experimental results. Section 5 discusses the findings and limitations. Lastly, Section 6 concludes this paper and gives future direction.

## 2. Motor Synergies

Humans can generate coordinated, efficient, and sophisticated movements by fully utilizing the dynamics of their complex musculoskeletal systems. The need to control and perpetually adjust a large number of DOFs is expected to be computationally formidable. To manipulate the inherently redundant musculoskeletal system for voluntary movement generation, the CNS must simultaneously activate and coordinate many muscles, each comprising thousands of motor units. Neuroscience researchers have proven that the concept of motor synergies exists within the CNS [25,26,27], whereby it simplifies control by combining several DOFs into synergies and significantly reducing the burden on the CNS.

There are numerous interpretations of this concept; most commonly, the CNS uses a considerably small set of instructions to control a large group of muscles for movement generation. Owing to the co-activation of a set of muscles using fewer neural commands, i.e., motor synergies, the EMG activities of these muscles tend to be spatially and temporally correlated, which is referred to as muscle synergies. At the same time, this covariation of the muscle activation imparts a certain level of coordination among closely related joints, coupling the angular movements in various joints, referred to as kinematic synergies [28]. Having excess DOFs allows the CNS the flexibility to use only those DOFs that align well with the task demands [29], enabling sophisticated and synergistic motion generation. This may explain the ability of humans to naturally perform complex movements in an energy-efficient manner without too much effort. Moreover, it also suggests that using a synergistic approach for motion estimation may allow more natural movement generation.

In various investigations, a wide range of motor behaviors has been explained and are suggested to be produced by synergies. The comprehensive analysis in [30] by d’Avella characterized the muscle synergies of healthy arm movements for a variety of upper limb reaching tasks spanning various directions, such as horizontal, up, and frontal. The original EMG envelopes for these highly variable upper limb movements were reconstructed using a reduced set of muscle synergies, which were observed to be repeatable and shared across directions and subjects. In the case of prosthesis control, this may be useful for developing a generalized and transferable control model using data from able-bodied subjects. The authors in [31] analyzed the kinematic synergies of various arm postures during unrestrained and natural catching movements, where the subjects were tasked to catch a ball thrown toward them along 16 different trajectories. The results show that 3 synergies were sufficient to represent approximately 90% of the variance in the recorded data for 10 joint angles, corresponding to 7 DOFs of the arm and 3 DOFs of the shoulder girdle. The analysis in [32] also featured similar results using joint angular velocities during reaching motions. These results justify that fewer kinematic synergies, with certain losses in accuracy depending on the number of synergies, can be used to represent and reconstruct the natural reaching movements of the human arm.

Many studies have identified synergies (i.e., motor primitives) at the electromyographic, kinetic, and kinematic levels during the last few decades. The calculations for extracting various types of synergies have been formalized by authors in [33,34,35,36]. In the present study, we used spatial synergies extracted from the joint angular movement data of the arm. The idea of inter-joint coordination during reaching movements of the human arm reflects that a set of DOFs potentially shows instantaneous covariations. Such types of movement primitives, implying the assumption that the ratios of the signals characterizing different DOFs remain constant over time [33], are referred to as spatial synergies. The spatial synergy decomposition is performed as in (Equation 1), where the source signals are represented by $x^{l}\left(t\right)$ at a time point *t* in trial number *l*, and *N* is the number of extracted spatial synergy components. $w_{n}$ indicates the spatial pattern of the kinematic synergies that are assumed to be invariant over the trials, whereas $c_{n}^{l}\left(t\right)$ are the corresponding activation signals that vary for each trial. We can rewrite (Equation 1) in a simplified matrix form as (Equation 2) after ignoring the *residuals* term, where *X* represents the source signals, *W* defines the synergy components, and *C* is the corresponding activation signal matrix.
(1)$$x^{l}\left(t\right)=\sum_{n=1}^{N}w_{n}\cdot c_{n}^{l}\left(t\right)+residuals$$
(2)X=W·C

For the analysis, we used the principal component analysis (PCA) method, which is one of the most widely used algorithms for solving (Equation 2); its core idea is to minimize the reconstruction errors *E* in (Equation 3) with respect to *W* and *C*, where ${∥\cdot ∥}_{F}$ indicates the Frobenius norm.
(3)$$E^{2}={∥X-W\cdot C∥}_{F}^{2}$$

## 3. Materials and Methods

### 3.1. Data Acquisition

Fourteen healthy right-handed subjects (thirteen males and one female) volunteered for this investigation. The subjects were 20–28 years old, with no known upper limb neuromuscular disorders. All subjects had given informed consent prior to participation in this experiment.

To record the motion data of a subject’s arm movements during reaching tasks, we used Perception neuron pro, an inertial measurement unit (IMU) sensor-based full-body motion capture system. Although the accuracy of this system is inferior to that of optical cameras, it is possible to capture motions without spatial constraints from anywhere within the communicable range of the device. The device uses individual sensors called neurons, each housing an IMU, attached to different body parts (Figure 2a). We used a total of eight sensors. Figure 2b shows the placement of each neuron—three on each arm (placed on the forearm, upper arm, and shoulder) and one each on the chest and lower back. The subjects can quickly wear the sensors using straps so that no additional preparations, such as special clothes, are necessary.

The sensor system communicates with the axis neuron pro software that processes raw motion data to formulate a 3D skeletal model in real time. Motion information from the developed skeletal model, such as the position and angle of each joint, can be obtained at a sampling frequency of 120 Hz. In our experiments, the data under consideration were the three shoulder orientation angles along X, Y, and Z axes, i.e., internal–external rotation, flexion–extension, and abduction–adduction of the shoulder joint, and two forearm orientation angles, i.e., pronation–supination and flexion–extension of the elbow joint; hereafter, these are referred to as $SH\theta_{x}$, $SH\theta_{y}$, and $SH\theta_{z}$ and $FA\theta_{x}$ and $FA\theta_{y}$, respectively.

### 3.2. Experimental Protocol

Bearing in mind that the purpose of a prosthetic device is to assist the user with their ADLs, we designed the workspace based on a routine activity by considering the user’s everyday environment instead of constrained movements in a laboratory or restricted environment. Reaching for objects placed on a table is a common scenario in daily life, which also targets arm movements in the horizontal plane only, specifically on the top surface of a table. A 40 cm × 40 cm target grid (Figure 3a) with a start/rest point and 8 numbered points was placed in a horizontal position on the surface of a table. The target numbered point to be reached was projected on the screen in front of the subject. The timing and color of the displayed numbers were controlled automatically to produce more consistent and regular movements, based on which the subjects had to perform the required reaching tasks. The subjects were provided instructions at the beginning of the experiments and allowed time to familiarize themselves with the environment. Therefore, no verbal commands or communications were required during the experiments, thus making the procedures easier to follow. Eight healthy subjects participated in these behavioral experiments. They were tasked with performing reaching movements to explore the top surface of a table placed before them while sitting straight on a chair, as shown in Figure 3b. All the subjects performed two sets of tasks, and kinematic data were acquired for the reaching movements of only the right arm as follows:

Dataset 1: with multiple (15 times) repetitive reaching movements toward each target point.Dataset 2: with reaching movements toward random target points (35 movements in total).

Dataset 1 with multiple repetitive movements was first used to extract the kinematic synergies and later as the training data for the neural network. Afterward, dataset 2, with reaching movements toward random target points, was used to cross-validate the trained neural network.

### 3.3. Kinematic Synergy Extraction

PCA has been used in numerous studies to investigate natural movements, such as catching [31] or reaching [32] tasks, where it was successful in representing the observed physiological complexities using fewer numbers of principal components (PCs) or synergies. One reason for this is that when capturing variances, the PCA uses a more intuitive method to exploit the coupling of the DOFs. Another important reason is that it considers linear correlations among the DOFs, which can be regarded as the minimal model of inter-joint coupling, i.e., a linear approach to explaining complex behaviors.

In this study, we performed a synergistic analysis using PCA as we are working with kinematic synergies. Dataset 1 was first segmented to acquire the data of interest, i.e., one individual reaching movement from the onset of the reaching motion until returning to the start/rest point (as in Figure 4). We then averaged the segmented data over the 15 trials for the same reaching movement, e.g., for target point 7. The averaged data were then low-pass filtered using a sixth-order Butterworth filter with a cutoff frequency of 10 Hz to remove motion artifacts and finally normalized to translate the angular values within the range of −1 to +1. For each subject, this averaged, filtered, and normalized dataset X (Equation 4) was obtained comprising submatrices $x_{j}^{m}\left(t_{max}\right)$ with joint angular values during reaching motions. Here, m=8 represents the total number of target points for the reaching tasks, j=5 is the number of DOFs or joint angles under consideration, and $t_{max}$ refers to a particular sample time for which the joints’ angular values were obtained. The *X* having size $j\times (m*t_{max})$ is fed to the PCA algorithm, which then provides the PCs. Each PC is a synergy representing the covariation of the joint angular configurations. The total number of PCs or synergy components “N” (Equation 1) is equal to the number of DOFs or joint angles under consideration, which is equal to five here. Figure 5a represents the extracted kinematic synergy component matrix (graphical representation) for one of the subjects.
(4)$$X=\left[\begin{array}{ccccc} x_{1}^{1}\left(t_{max}\right) & \cdots & x_{1}^{4}\left(t_{max}\right) & \cdots & x_{1}^{8}\left(t_{max}\right)\\ x_{2}^{1}\left(t_{max}\right) & \cdots & x_{2}^{4}\left(t_{max}\right) & \cdots & x_{2}^{8}\left(t_{max}\right)\\ \vdots & \ddots & \vdots & \ddots & \vdots \\ x_{5}^{1}\left(t_{max}\right) & \cdots & x_{5}^{4}\left(t_{max}\right) & \cdots & x_{5}^{8}\left(t_{max}\right) \end{array}\right]$$

We retained the minimum of the most significant PCs that explain at least >85% of the total variance. Figure 5b presents the variations of the PCs for subject 1. We can observe that the first synergy accounts for more than 75% of the variance, and the sum of the first two synergies can account for more than 90% of the overall variance in the source data. Following our set criteria, it suggests that we can adequately approximate the original data using only the first two synergies that capture a large portion of the variance. Thus, the subject’s original movements can be reconstructed with an acceptable loss in accuracy. Figure 4 represents the difference in the reconstructed data for one of the subjects when represented using all five synergies (shown in red) and when using only two synergies (shown in green). It can be seen that the red curve is an exact match for the original movement data (shown in black). In contrast, the green curve still represents the original data reasonably accurately. The last two rows show the activation signals C (in blue) in the case of two synergies, i.e., C1 and C2. Based on these observations, using the synergy-space neural network approach, it is plausible to use only the first two synergies to estimate the forearm motions.

### 3.4. LSTM Training

In the present work, we trained a neural network to predict the extracted activation signals based on shoulder kinematics. As both sets of data are time-series signals, it is necessary to use an ANN that suitably processes the time-series or sequential data. Different ANN architectures have been employed in various studies to determine inter-joint coordination during human arm movements. The authors in [37] used a radial basis function network (RBFN)-based neural network, whereas [38] used a time-delayed adaptive neural network (TDANN) to estimate the distal joint angles. In this study, we used LSTM, a particular type of RNN capable of handling long-term dependencies. LSTMs have internal mechanisms called gates that regulate the flow of information to handle the vanishing gradient problem in RNNs, thus making them very suitable for multivariate time-series forecasting.

We used python’s machine learning library, Keras, to implement the LSTM model. There are various parameters in the neural network, and the estimation accuracy may change depending on the settings of these parameters. Therefore, we first tested the learning efficiency and estimation accuracy of the LSTM model by varying the parameters, such as the batch input size, number of LSTM hidden layers, number of nodes in each layer, and number of training epochs. These parameters affect the learning and estimation efficiencies of the model. The batch size controls the number of samples shown to the network before the weight updates are applied. If the input batch size is large, the model can quickly process the entire training dataset; however, it can overlook certain features during training that might be crucial to learning.

On the other hand, we can increase the complexity and expressiveness of the model by optimizing the number of LSTM hidden layers and the number of neurons in each layer. Even in the case of using only a single hidden layer, the LSTM model can learn the characteristics of the time-series data. The efficiency can be improved by stacking multiple layers. However, if the model is made more complex than necessary, the training may not be effective. Similarly, if we increase the number of nodes, the model requires more time for learning with no significant change in accuracy. The same is also true for the number of training epochs. Estimating the time-varying activation signals is classified as a regression problem, so we use the mean squared error (MSE) as the loss function and Adam as the optimization function, as has been widely used in similar studies.

Lastly, to avoid the overfitting problem, a dropout rate of 10% is used in each layer, whereby 10% of the neurons are dropped randomly. Supervised learning is then carried out, where the neural network develops the regression model based on the input–output pairs. As for the training data, at a single time step, we can apply a total of six inputs to the model: the $SH\theta_{x}$, $SH\theta_{y}$, and $SH\theta_{z}$ angle of the shoulder joint and their respective derivatives $SH{\dot{\theta }}_{x}$, $SH{\dot{\theta }}_{y}$, and $SH{\dot{\theta }}_{z}$ (i.e., shoulder joint angular velocities). It is also possible to input multiple time-step data to the LSTM model at a time using the last few time-step data to predict the output for the current time step. This can improve the estimation accuracy at the cost of increasing calculations. We used the ten previous time-step data as inputs to the model (Figure 6). The model outputs were the activation signals, which in the case of two synergies are $C1_{p}$ and $C2_{p}$.

### 3.5. Analysis Strategy

In the present work, we compared the performances of the synergy-space neural network approach, where the shoulder kinematics are mapped to synergistic activation signals, with the direct estimation approach, wherein the neural network is used to map the shoulder kinematics to the forearm kinematics. To thoroughly investigate this comparison, we devised a comprehensive strategy to train and test 36 different LSTM models for each subject and analyze the performances. The devised strategy was based on the following criteria.

#### 3.5.1. Learning Methodology

First, we devised a scenario based on our proposed learning methodology for the LSTM model. This defines the approach chosen for training the network, that is, either synergy-space or direct estimation. Three strategies were devised based on the number of synergistic components used and the learning approach.

Synergy-Space estimation using 2 synergies components:In this case, we first extract the synergies *W* and their corresponding activation signals *C* from the reaching motion data considering five DOFs of the arm, i.e., $SH\theta_{x}$, $SH\theta_{y}$, $SH\theta_{z}$, $FA\theta_{x}$, and $FA\theta_{y}$. Subsequently, we train the LSTM model to predict two activation signals C1 and C2 based on the shoulder kinematics provided as inputs. The predicted activation signals $C1_{p}$ and $C2_{p}$ are then used along with the synergy matrix *W* to estimate the required forearm motions.Synergy-space estimation using 1 synergy component:In this case, we extracted the synergies *W* and their corresponding activation signals *C* from the reaching motion data considering only two DOFs of the forearm, i.e., $FA\theta_{x}$ and $FA\theta_{y}$. We then trained the LSTM model to predict only one of the activation signals C1, providing the shoulder kinematics as the input. Finally, the predicted activation signal $C1_{p}$ and the synergy matrix *W* are used to estimate the required forearm motions.Direct estimation:For direct estimation, no prior information or synergistic components are extracted from the reaching motion data recorded during arm movements. The LSTM model is directly trained to predict $FA\theta_{x}$ and $FA\theta_{y}$ angles of the elbow joint based on the input shoulder kinematics.

#### 3.5.2. Number of LSTM Hidden Layers

As noted previously, a single layer of LSTM can learn the necessary features of the time-series data. By stacking multiple layers of LSTMs, this ability can be enhanced. Thus, to verify the appropriate number of hidden layers for our task, we tested three scenarios with varying numbers of hidden LSTM layers in our model. For each learning methodology mentioned earlier, we constructed and trained three different models, namely M1, M2, and M3, having one, two, and three hidden LSTM layers, respectively.

#### 3.5.3. Number of Inputs

As previously mentioned, we can apply a total of six inputs to our model, represented by $SH\theta_{x}$, $SH\theta_{y}$, $SH\theta_{z}$, $SH{\dot{\theta }}_{x}$, $SH{\dot{\theta }}_{y}$, and $SH{\dot{\theta }}_{z}$. However, various combinations of these signals can also be used as inputs. This consideration was based on the fact that the level of residual limb movement control would vary for the user depending on the severity of the amputation. In addition, wearing a prosthetic socket can limit the range and types of movements the user can perform. In many cases, the shoulder internal rotation motion is the most difficult to perform for amputees. The other reason was to test whether there is any advantage to using shoulder joint angular velocities as the inputs. We created four different training scenarios with different numbers of inputs to the LSTM models. The combined and total numbers of inputs for each scenario are shown in Table 1.

### 3.6. Evaluation

#### 3.6.1. RMSE

For the evaluation of the trained LSTM model, the estimated forearm orientation angles (i.e., pronation–supination $FA\theta_{x}$ and flexion–extension $FA\theta_{y}$ angles of the elbow joint) were compared to the actual forearm orientation angles captured using the neuron pro system. Figure 7 presents a sample of the continuous signal plot comparing the actual vs. estimated forearm motions during one of the scenarios tested for subject 5. To assess the performance of the joint angle estimations, the standard metric used is the root mean squared error (RMSE) [38] as given in Equation (Equation 5), where ${\hat{x}}_{t}$ is the predicted joint angle, $x_{t}$ is the actual joint angle at data point *t*, and *N* is the total number of data points.

To calculate the RMSE value, we used the “mean_squared_error” metric from the scikit-learn library for python and applied the square root. In the case of multiple outputs, this metric gives an average value of the RMSE.
(5)$$\mathrm{RMSE}=\sqrt{\frac{1}{N}\sum_{t=0}^{N}{\left({\hat{x}}_{t}-x_{t}\right)}^{2}}$$

#### 3.6.2. Pearson Correlation Coefficient

Pearson’s correlation method analyzes the linear relationship between two variables and provides a coefficient value as a measure of the correlation strength. The Pearson correlation coefficient is denoted by *r* and can have a value between +1 and −1. Table 2 presents the detailed interpretation of the Pearson correlation coefficient.

We used the “corrcoef” function from python’s NumPy library, which uses the actual and estimated values of forearm orientation angles (as in Figure 7) to compute the Pearson correlation coefficient.

#### 3.6.3. Analysis of Variance (ANOVA) Test

To statistically verify the differences between the results obtained using the synergy-space neural network and direct estimation methods, we performed the analysis of variance (ANOVA) test. The ANOVA compares three or more populations to ascertain whether the variability between group means is larger than the variability in the observations within the groups. A significance level or threshold is chosen, and a *p*-value less than the threshold is interpreted as evidence of the difference between the population means. In this study, the *p*-value indicates significant differences between the learning strategies’ results.

To show the pertinence of each learning methodology, we perform the ANOVA test using the RMSE values obtained by comparing the estimated and actual forearm motions for the various LSTM models trained using the synergy-space and direct estimation approaches. We chose a significance level of 0.05, one of the standard choices. Suppose the calculated *p*-value is less than the threshold. In that case, the statistically significant ANOVA is followed up with the Tukey HSD (honest significant difference), a post hoc test pinpointing which learning methodology exhibits a statistically significant difference.

#### 3.6.4. Cross-Subject Analysis

As discussed previously, one of the characteristics of the synergies is that they are shared among similar tasks to some extent. This suggests that a generalized or transferable control model can be developed based on the synergy-space neural network approach using the data recorded from healthy subjects. Therefore, cross-subject testing was performed to test this assumption for the robustness of the learning methodologies.

For the case of the synergy-space method, we performed the cross-subject analysis using, for example, subject A’s input data fed to the LSTM models trained using the other subjects’ data and then employing subject A’s synergy matrix for the forearm motion prediction. However, for the case of the direct estimation method, the cross-subject analysis was performed using, for example, subject A’s input data fed to the LSTM models trained using the other subjects’ data for subject A’s forearm motion prediction.

The evaluations were performed by calculating the RMSE values to compare the estimated motions with the actual measured values. For the cross-subject analysis, we used model M2 with six inputs (i.e., best-case scenario) and model M2 with two inputs (i.e., worst-case scenario). The ANOVA was then performed to statistically verify the difference between the cross-subject results based on the synergy-space and direct estimation approaches.

## 4. Results

In this study, we trained and evaluated 36 different LSTM models for each of the 14 subjects. The training strategy was devised based on three different learning methodologies. We constructed 3 separate LSTM models and trained each model using 4 different combinations of the input signals (i.e., 3×3×4=36).

The acquired RMSE values of the estimated forearm motions ($FA\theta_{x}$ and $FA\theta_{y}$) for some subjects are shown as bar plots in Figure 8. It was observed that, overall, there are no significant differences in the results based on the learning methodology. This suggests that even with the reduced state-space representation, the synergy-space neural network is capable of keeping the performance similar to that of the direct estimation method. However, we see notable differences when using combinations of two and three inputs, implying that shoulder internal–external rotations ($SH\theta_{z}$) as the input significantly increase the estimation accuracy, which is the case in all the training scenarios. On the other hand, the joint angular velocities as the inputs provided a minor increase in the accuracy, as is visible when comparing the RMSE values based on the number of inputs 2 and 4 and also 3 and 6. However, no marked differences were observed in the performance based on the number of hidden LSTM layers among the models M1, M2, and M3. These observations are valid for all the subjects included in the study.

Regarding the minor differences for all the trained scenarios for each subject in the study, we obtained three general result types. First, similar to subject 1 (Figure 8), the synergy-space method (using 2 synergies) produced the lowest RMSE values with an overall RMSE and standard deviation value of $10.88^{\circ }$ and $2.09^{\circ }$, respectively, for all the tested scenarios. Second, similar to subject 2 (Figure 8), we obtained almost similar RMSE values having some variations from an overall RMSE and standard deviation value of $7.55^{\circ }$ and $1.16^{\circ }$, respectively, for all the learning methodologies. Lastly, similar to subject 8 (Figure 8), the direct estimation method had the lowest RMSE values with an overall RMSE and standard deviation value of $3.99^{\circ }$ and $1.47^{\circ }$, respectively, for all the tested scenarios. We only selected the scenarios trained using model M2 for further analyses, i.e., with two hidden LSTM layers, as it produced the best overall results.

### 4.1. Personalized LSTM Models Evaluation

To evaluate the personalized LSTM models and compare the different learning methodologies, we used the LSTM model M2, which has two hidden LSTM layers and six inputs (Table 1). The average RMSE values obtained for the estimated $FA\theta_{x}$ and $FA\theta_{y}$ were $4.24^{\circ }$ and $9.75^{\circ }$, $6.08^{\circ }$ and $9.86^{\circ }$, and $5.84^{\circ }$ and $7.40^{\circ }$ for the two-synergy, one-synergy, and direct estimation methods, respectively.

We first employed Pearson’s correlation coefficient to quantify the estimation performance of the different learning methodologies. Table 3 shows a mostly strong positive linear correlation (+0.7≤r≥+1.0) for the personalized LSTM models. Furthermore, even for the few cases of a distinct or weak linear relation (highlighted cells in Table 3), the trend is similar for all the learning methodologies, suggesting a similar overall performance.

To show the pertinence of each learning methodology, we calculated the *p*-value using the ANOVA between the results of the training strategies using the RMSE values of the estimated forearm motions. The statistical ANOVA test associated with the RMSE values obtained for the three learning methodologies is reported in Table 4. The ANOVA provided evidence that there was no statistically significant difference, F(2,39)=1.705,p=0.195. The summary of the ANOVA test results is shown in Table 5. As the *p*-value corresponding to the F Statistic is greater than the threshold value of 0.05, this is interpreted as there is no significant difference between the population means and eliminates the need to perform any post hoc or multiple comparison corrections test, such as the Tukey HSD. Thus, we can say that the three training strategies tend to produce similar RMSE values, and there is no significant difference between their performances for subject-specific or personalized LSTM models. This further validates that the proposed synergy-space approach for mapping the inter-joint coordination for each subject’s personalized ANN models performs on par with the direct estimation method even with reduced dimensionality.

### 4.2. Cross-Subject Evaluation

With transhumeral amputees, the target application of this study, it is impossible to measure the elbow joint or forearm motion information. Therefore, testing the transferability of the trained predictive models is crucial.

Figure 9 shows a box plot summarizing the RMSE values for each subject’s cross-subject evaluations using the LSTM model M2 with six inputs. The box size reflects the range where 75% of the sample values lie, with a smaller box size indicating less variation in the estimation performance. The results demonstrate that the synergy-space neural network, particularly when using two synergies, exhibits stronger robustness to inter-individual variability compared to the direct estimation method. This may be attributed to the shared nature of the synergies across similar tasks and subjects, which enables the network to learn features common to the human arm’s reaching task. The descriptive statistics associated with the RMSE values obtained for the cross-subject testing are reported in Table 6, where the proposed synergy-space RNN exhibited an average reduction of 50% in the variation in the RMSE compared to the direct estimation method (highlighted cells in Table 6). These results demonstrate the effectiveness of the proposed synergy-space RNN in achieving better transferability during the cross-subject evaluation.

To statistically verify whether the difference in learning methodologies affected the performance during the cross-subject evaluations, we again calculated the *p*-values using the ANOVA between the results using the RMSE values of the estimated forearm motions. The ANOVA yielded a statistically significant effect, F(2,585)=3.227,p=0.040. Table 7 shows the summary of the ANOVA test results. As the *p*-value corresponding to the F Statistic is lower than the threshold value of 0.05, this suggests that the performance of one or more learning methodologies is significantly different. We further evaluated the nature of the differences between the three population means, i.e., to check which learning methodology tends to perform differently from the others.

The statistically significant ANOVA was followed-up with the Tukey HSD (honest significant difference), a post hoc test pinpointing which learning methodology exhibits a statistically significant difference. The post hoc Tukey HSD test results are reported in Table 8. The *p*-values corresponding to the Q Statistic are lower than 0.05 in the cases when comparing the direct estimation method with the synergy-space neural network approach (when using two synergies), suggesting a significant difference in the performance of the two approaches.

Similar results were also obtained for the cross-subject evaluations using the LSTM model M2 with two inputs only, presented in Table 9, Table 10 and Table 11. Even when using only two inputs (considering limited shoulder internal rotation motion (Table 1)) the proposed synergy-space method had about 40% less variation in the RMSE compared to the direct estimation method (highlighted cells in Table 9). The evaluation of the cross-subject analysis suggests that the synergy-space approach is more robust and may provide the possibility of developing a transferable model for prosthesis control.

## 5. Discussion

We have proposed and evaluated the synergy-space neural network for transhumeral prosthesis control. By explicitly incorporating kinematic synergies into the model, our approach addresses the limitations of traditional ANNs and provides a more robust and superior transferability across different subjects. Our rigorous evaluation of the model has shown promising results, demonstrating its potential.

We evaluated the performance of the proposed synergy-space approach for personalized LSTM models and compared it to the direct estimation method. The results of Pearson’s correlation method and the ANOVA analysis indicate that the proposed method performs comparably to the direct estimation method, with no significant difference in performance. However, the proposed approach still performs well, even with the reduction in dimensionality, suggesting its efficient and better learning capabilities for personalized LSTM models.

In the various tested scenarios, we observed that using shoulder internal–external rotations ($SH\theta_{x}$) as the input significantly increases the estimation accuracy of the LSTM models, which was typical for all the learning methodologies. It can be because the shoulder rotation is coupled to the forearm rotation; however, we have already extracted the kinematic synergies from the subjects’ arm motion data corresponding to the five DOFs of the arm, including the shoulder and forearm rotations (i.e., $SH\theta_{x}$ and $FA\theta_{x}$). It seems not associated with the joint coordination issue and is more concerned with learning the LSTM model. An additional input (i.e., $SH\theta_{x}$) provides an additional parameter to the LSTM model during the supervised training/learning, and probably a more unique feature compared to using the joint angular velocities (i.e., $SH{\dot{\theta }}_{y}$ and $SH{\dot{\theta }}_{z}$). Therefore, it improves the model’s accuracy.

The synergy-space approach demonstrated its superiority during the cross-subject evaluation as a more robust and transferable learning methodology. It showed more minor variations in the estimation accuracy when using one subject’s motion data and extracted the synergy matrix for forearm motion estimation using the personalized LSTM models of the other thirteen subjects. However, one of the limitations of this study for the actual implementation on amputee users will be obtaining the subject-specific synergy matrix. This is because, as mentioned earlier, amputee users cannot provide the necessary motion information.

Based on the properties of synergies being repeatable and shared across similar tasks and subjects, one practical solution can be to create a generalized synergy matrix based on the data of all the able-bodied participants and use it for amputee users’ forearm motion prediction. Another possibility would be to use motion data from the user’s healthy arm to generate a synergy matrix for the amputated arm. As the synergy matrix represents the inter-joint coordination, the LSTM model needs to learn a simplified relation between the shoulder kinematics and the activation signals. That means effectively extracting synergies can significantly affect performance.

This first investigation of the proposed synergy-space neural network demonstrates its potential as a robust and transferable predictive model, which was successfully confirmed through the cross-subject evaluation results. This finding can contribute toward creating a synergistic and generalized control strategy for transhumeral prostheses and other rehabilitation applications.

## 6. Conclusions

The primary aim of this study was to improve the control of transhumeral prostheses, focusing on their transferability across users. A highly accurate transferable predictive model is necessary for transhumeral amputees because individual calibration or personalized learning methods cannot be used effectively as they cannot provide the required data.

In this research, we proposed the synergy-space neural networks, as a transferable model, to predict the joint angles of the forearm motion based on the residual shoulder motion. We presented the implementation and evaluation of the proposed method, discussing its learnability and robustness for transferability to amputee users. The study was conducted with able-bodied subjects, focusing on reaching movements of the arm in the horizontal plane only. We compared the synergy-space neural network approach with the direct estimation method, using the actual and estimated joint angular values for the performance evaluation. In the best-case scenario, average RMSEs of $9.75^{\circ }$ and $4.24^{\circ }$ were achieved using the synergy-space method (using 2 synergies) for the flexion–extension ($FA\theta_{y}$) and pronation–supination ($FA\theta_{x}$) angles of the forearm motion. Consequently, we verified that for the case of personalized predictive models, even with a reduced state space, the proposed synergy-space neural network approach produced results similar to the direct estimation method.

We investigated the transfer learning ability of the proposed model through cross-subject analysis. The results indicate that the synergy-space neural network exhibited superior learning capabilities compared to the traditional direct estimation method during cross-subject evaluations. This highlights the strength of our approach as a transferable decoder, demonstrating its ability to handle inter-individual variabilities and providing a more generalized model for transhumeral prosthesis control.

In the future, we can send the output of predicted joint angles from the proposed model to a transhumeral prosthesis for real-time control. As a next step, this approach would be extended to incorporate reaching motions in three-dimensional space. The ultimate goal in the future is to develop a framework for the real-time estimation of forearm motions to further test and improve the proposed approach, such that it can be employed on actual transhumeral prostheses that allow users to control the device intuitively.

## Figures and Tables

**Figure 1 sensors-23-04188-f001:**
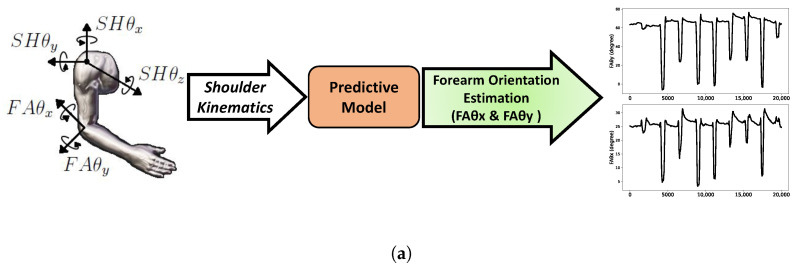
Illustration of the idea of this study: (**a**) a simplistic representation of direct estimation method; (**b**) the proposed synergy-space neural network method. Here, $SH\theta_{x}$, $SH\theta_{y}$, and $SH\theta_{z}$ are the shoulder kinematics, and $FA\theta_{x}$ and $FA\theta_{y}$ represent the forearm orientations. The PCA blocks symbolize the process of synergy extraction, where *W* represents the synergistic components and *C* is the corresponding activation signal matrix. The predictive model is the trained RNN that outputs the predicted activation signals $C_{p}$, with the cross-operator representing the matrix multiplication of the incoming values.

**Figure 2 sensors-23-04188-f002:**
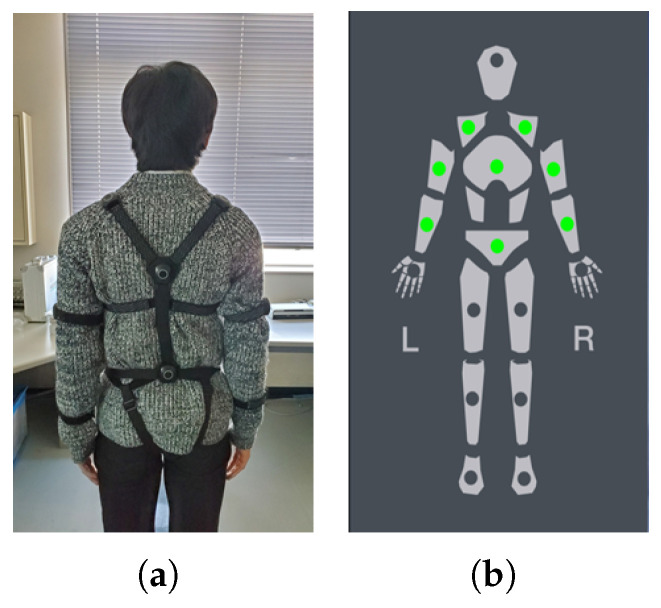
Neuron pro sensor placements for capturing the required motion data: (**a**) subjects wearing neuron sensors; (**b**) green spots marking the placements of the sensors for the upper body mode of neuron pro because only arm movement data are required.

**Figure 3 sensors-23-04188-f003:**
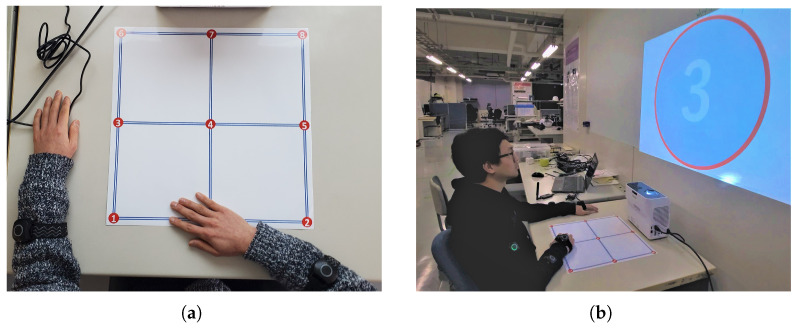
Experimental setup for the target reaching tasks: (**a**) target grid with a subject’s right hand at the start point; (**b**) complete experimental setup for the target reaching tasks.

**Figure 4 sensors-23-04188-f004:**
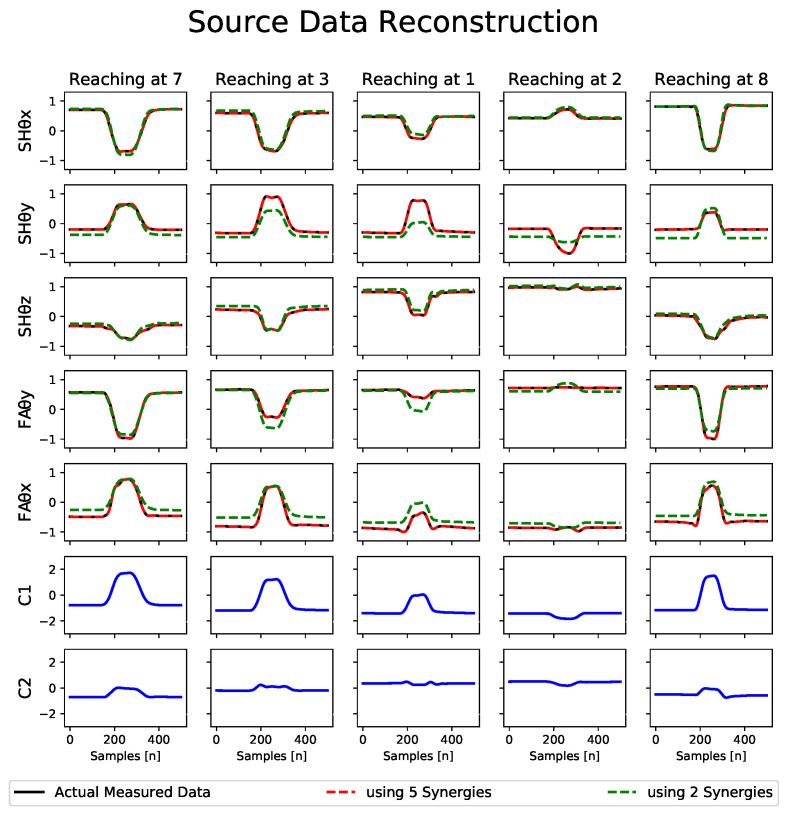
The first 5 rows represent the shoulder and elbow joints’ angular values normalized between −1 and +1. It comprises the source data (solid black curves) of subject one and its reconstructions using all five synergies (red dotted curves) and only two synergies (green dotted curves). The last two rows represent the corresponding activation signals C (C1 and C2 in the case of two synergies).

**Figure 5 sensors-23-04188-f005:**
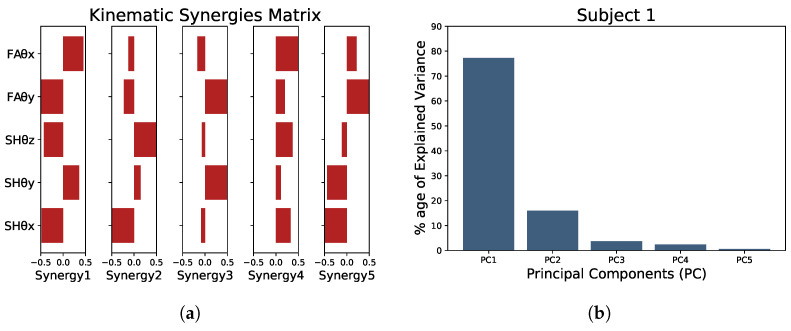
Example of the kinematic synergies of subject 1. (**a**) The spatial synergies extracted from the training data. The notations $SH\theta_{x}$, $SH\theta_{y}$, $SH\theta_{z}$, $FA\theta_{x}$, and $FA\theta_{y}$ indicate the axis of the degree of freedom. (**b**) Bar plot showing the importance of each principal component in explaining the variance in the source data.

**Figure 6 sensors-23-04188-f006:**
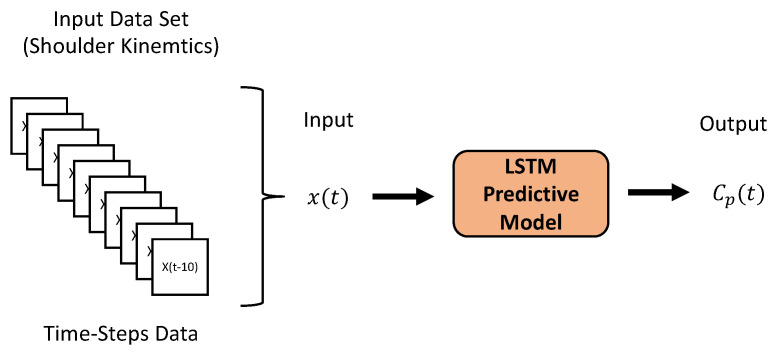
Input dataset creation: 10 previous time-steps data are combined and provided as the input x(t) to the LSTM model to predict the output $C_{p}\left(t\right)$ at the current time step *t*.

**Figure 7 sensors-23-04188-f007:**
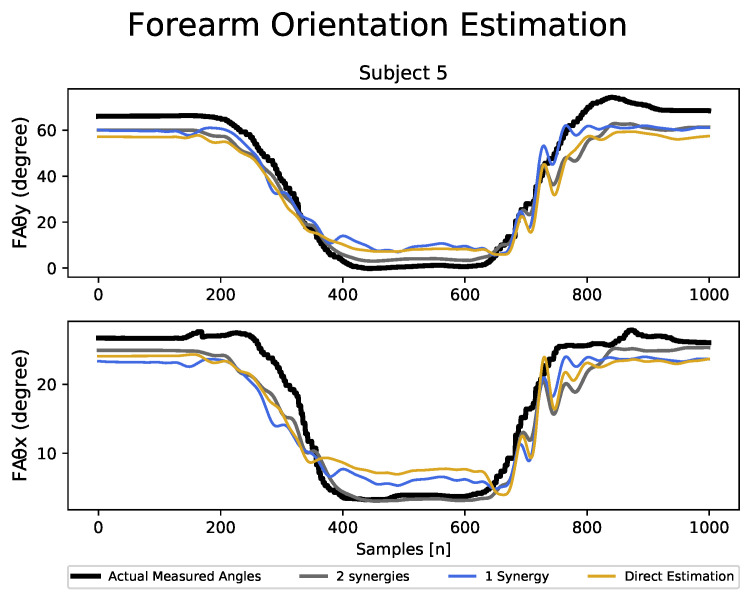
Sample of the joint angular value plots of the actual vs. estimated forearm motions using two synergies, one synergy, and direct estimation learning methodologies for one of the tested scenarios of subject 5.

**Figure 8 sensors-23-04188-f008:**
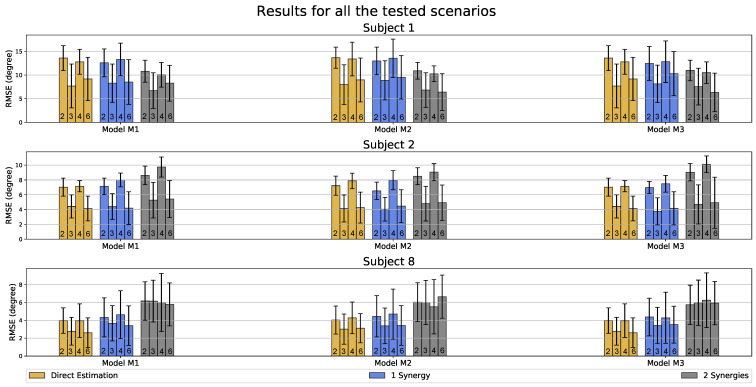
Results of the forearm motion estimations ($FA\theta_{x}$ and $FA\theta_{y}$) for all tested scenarios for subjects 1, 2, and 8, where each bar represents the calculated RMSE. The bars are first divided into M1, M2, and M3 based on the number of LSTM hidden layers in the model. The golden, blue, and gray bars represent the errors in the estimations when using two synergies, one synergy, and direct estimation learning approaches, respectively. The number inside the bar represents the number of inputs to the LSTM model (Table 1), whereas the error bar represents the standard deviation of estimation error values.

**Figure 9 sensors-23-04188-f009:**
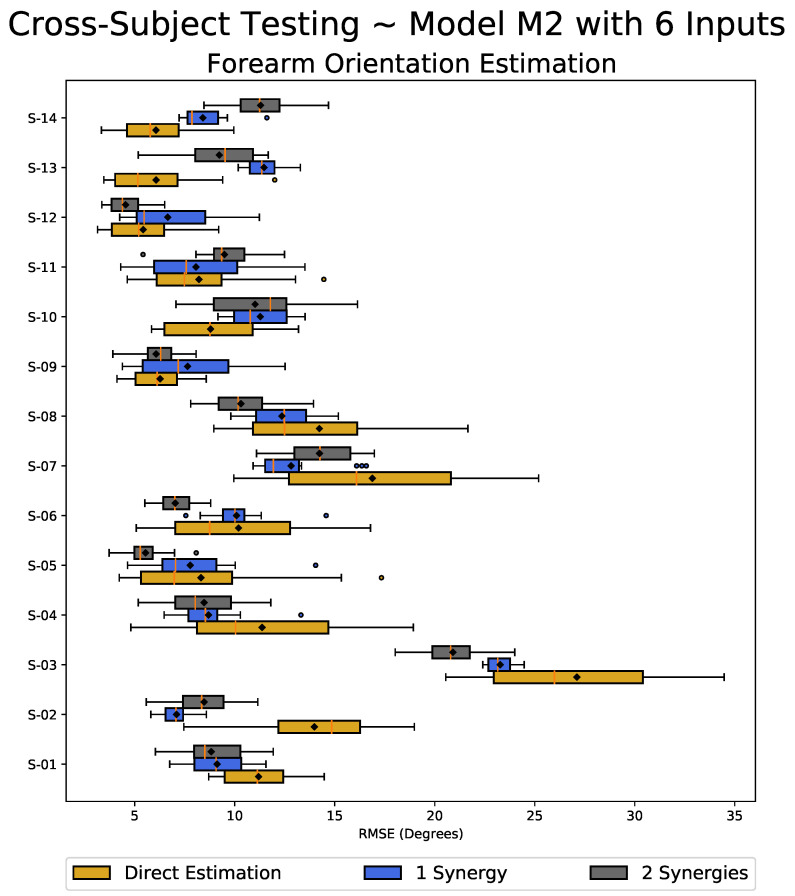
The box plot shows RMSE values for each subject’s cross-subject evaluation. The box size represents the range of 75% of the values, and the solid vertical golden line inside the box represents the median, with the black diamond marker indicating the mean value. Outliers are represented by circular markers, and the whiskers show the maximum and minimum values. A smaller box size represents minor variation in results and better transferability.

**Table 1 sensors-23-04188-t001:** Training Scenarios based on No. of Inputs.

No. of Inputs	Signal Combination					
	$\mathrm{SH}\theta_{x}$	$\mathrm{SH}\theta_{y}$	$\mathrm{SH}\theta_{z}$	$\mathrm{SH}{\dot{\theta }}_{x}$	$\mathrm{SH}{\dot{\theta }}_{y}$	$\mathrm{SH}{\dot{\theta }}_{z}$
2		✔	✔			
3	✔	✔	✔			
4		✔	✔		✔	✔
6	✔	✔	✔	✔	✔	✔

Note: ✔ cell means signal used as input.

**Table 2 sensors-23-04188-t002:** Interpretation of Pearson Correlation Coefficient.

Range of *r*	Degree of Relationship
−1.0≤r≥−0.7	A strong negative linear relationship
−0.7≤r≥−0.3	A distinct negative linear relationship
−0.3≤r≥−0.1	A weak negative linear relationship
−0.1≤r≥+0.1	Not a linear relationship
+0.1≤r≥+0.3	A weak positive linear relationship
+0.3≤r≥+0.7	A distinct positive linear relationship
+0.7≤r≥+1.0	A strong positive linear relationship

**Table 3 sensors-23-04188-t003:** Pearson’s correlation coefficients “*r*” for Model M2 with 6 inputs.

Subject No.	Learning Methodology					
	Direct Estimation		1 Synergy		2 Synergies	
	FE	PS	FE	PS	FE	PS
1	0.91	0.75	0.91	0.78	0.94	0.75
2	0.95	0.89	0.92	0.89	0.96	0.90
3	0.96	0.60	0.89	0.86	0.92	0.70
4	0.96	0.71	0.91	0.70	0.97	0.71
5	0.98	0.90	0.98	0.89	0.98	0.92
6	0.97	0.83	0.93	0.76	0.95	0.84
7	0.90	0.86	0.89	0.86	0.92	0.89
8	0.97	0.38	0.93	0.67	0.96	0.51
9	0.94	0.63	0.96	0.51	0.94	0.57
10	0.91	0.31	0.92	0.29	0.88	0.34
11	0.93	0.56	0.96	0.28	0.94	0.45
12	0.98	0.80	0.98	0.83	0.97	0.81
13	0.96	0.35	0.92	0.30	0.94	0.42
14	0.96	0.70	0.92	0.57	0.95	0.65

Note: FE = flexion–extension, PS = pronation–supination. Highlighted cells mark weak correlation.

**Table 4 sensors-23-04188-t004:** Descriptive statistics of RMSE values obtained for different learning methodologies using model M2 scenarios only.

Learning Methodology	Count	Sum	Average	Variance
Direct Estimation	14	86.756	6.197	4.672
1 Synergy	14	107.930	7.709	7.335
2 Synergies	14	108.972	7.784	7.748

**Table 5 sensors-23-04188-t005:** ANOVA summary table for the results using model M2 scenarios only.

Source of Variation	SS	df	MS	F Statistic	*p*-Value	F Critical
Between Methodologies	22.452	2	11.226	1.705	0.195	3.238
Within Methodology	256.826	39	6.585			
Total	279.278	41				

Note: SS = Sum of Squares, df = Degrees of Freedom, MS = Mean Square.

**Table 6 sensors-23-04188-t006:** Descriptive statistics of the RMSE values obtained for different learning methodologies during cross-subject evaluation using model M2 with 6 inputs.

Learning Methodology	Count	Sum	Average	Variance
Direct Estimation	196	2157.122	11.006	42.696
1 Synergy	196	2025.910	10.336	19.789
2 Synergies	196	1895.985	9.673	18.365

Note: Highlighted cells mark the least values of average and variance.

**Table 7 sensors-23-04188-t007:** ANOVA summary table for the cross-subject evaluation using model M2 with 6 inputs.

Source of Variation	SS	df	MS	F Statistic	*p*-Value	F Critical
Between Methodologies	173.962	2	86.981	3.227	0.040	3.011
Within Methodology	15,765.779	585	26.950			
Total	15,939.741	587				

Note: SS = Sum of Squares, df = Degrees of Freedom, MS = Mean Square.

**Table 8 sensors-23-04188-t008:** Post hoc Tukey HSD test results for the cross-subject evaluation using model M2 with 6 inputs.

Group Pair	Q Statistic	*p*-Value	Q Critical
Direct Estimation vs. 1 Synergy	1.805	0.411	
Direct Estimation vs. 2 Synergies	3.593	0.030	3.323
1 Synergy vs. 2 Synergies	1.788	0.418	

**Table 9 sensors-23-04188-t009:** Descriptive statistics of the RMSE values obtained for different learning methodologies during cross-subject evaluation using model M2 with 2 inputs.

Learning Methodology	Count	Sum	Average	Variance
Direct Estimation	196	2792.539	14.248	41.462
1 Synergy	196	2646.628	13.503	24.561
2 Synergies	196	2440.454	12.451	23.557

Note: Highlighted cells mark the least values of average and variance.

**Table 10 sensors-23-04188-t010:** ANOVA summary table for the cross-subject evaluation using model M2 with 2 inputs.

Source of Variation	SS	df	MS	F Statistic	*p*-Value	F Critical
Between Methodologies	319.323	2	159.661	5.347	0.004	3.011
Within Methodology	17,468.049	585	29.860			
Total	17,787.372	587				

Note: SS = Sum of Squares, df = Degrees of Freedom, MS = Mean Square.

**Table 11 sensors-23-04188-t011:** Post hoc Tukey HSD test results for the cross-subject evaluation using model M2 with 2 inputs.

Group Pair	Q Statistic	*p*-Value	Q Critical
Direct Estimation vs. 1 Synergy	1.907	0.370	
Direct Estimation vs. 2 Synergies	4.602	0.003	3.323
1 Synergy vs. 2 Synergies	2.695	0.138	

## Data Availability

Not applicable.

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
