# Peer review of "Synergy-Space Recurrent Neural Network for Transferable Forearm Motion Prediction from Residual Limb Motion"

_sensors, 2023, doi:10.3390/s23094188_

Round 1

Reviewer 1 Report

The manuscript presents a machine learning model for predicting forearm motion from residual limb kinematics, with the goal of improving the control of transhumeral prosthetic devices for reaching tasks. The authors propose a novel synergy (shoulder-elbow kinematics coupling) based recurrent neural network, which enables transfer learning between different subjects. The authors conducted an extensive experimental evaluation of the proposed model, comparing it with a conventional direct estimation neural network method. The results show that the proposed model outperforms the direct estimation method on the transferability between subjects, demonstrating the effectiveness of the proposed approach for transferable forearm motion prediction. Overall, the manuscript is well-written and provides a clear description of the proposed model and its motivation. The authors provide a thorough analysis of the model's performance and compare it with the direct estimate method.

One potential limitation is the training and test of the direct estimation for cross-subject analysis. While the description of cross-subject analysis about the synergy space-based method is sufficiently detailed, that of the direct estimation is insufficient and unclear.

Is it simply that “one subject’s input data is fed to the LSTM trained with the other subjects‘ data”? If this is true, the comparison might be unfair, since for the proposed synergy space-based method, it could not only use one subject’s input data to the LSTM trained with the other subjects’ data, but also it can use this subject’s synergy data.

There is one possibility that, there are a limited number of (e.g., 2 or 3) different synergy patterns, which makes it difficult for the direct estimation network trained for one certain synergy pattern to match another synergy pattern.

In order to make this clear, the synergy patterns of the subjects need to be shown and analyzed.

Even if the synergy patterns are individual dependent and can’t be classified into 2 or 3 clusters, the readers can have an intuitive understanding of complexity of the synergy patterns.

If the assumption is true, in this case, if multiple direct estimation networks could be built accordingly and properly selected, or if multiple (a limited number of, say 2 to 3) synergy patterns could be embedded into the direct estimation neural networks, high prediction accuracy and transferability might be achieved, too. This should be investigated and discussed.

Some figures or the explanation of figures need to be improved.

Figure 1(b) is not clearly explained either in its caption and related paragraphs. Readers need to first read the subsequent paragraphs to get the meaning of some symbols C1, C2, C1p, C2p in the graph.

Similarly, in figure 6, the meaning of xt is not clear.

In Table 3, it is better to use a different color to mark items of weak correlation.

In Figure 8, please also have standard deviation expressed.

When explaining figure 8, in around line 371, it is better to use real values to describe and compare the results about different learning strategies.

Some minors

Line 390, rather than Table 1, it should be Table 3,

Line 190, what is the meaning of “??”

Line 337, The->the

Reviewer 2 Report

Minor issues:

1. Check for grammar mistakes. Occasionally, you use capital letters although you are inside a sentence.

2. Fig. 6 is low quality, please improve it.

Medium issues:

1. State or the art has less than 30 references. And at one point, you cite 6 references at a time. It seems superficial. Redo it.

2. The study presented here looks as an improvement of your previous study "Ahmed, Muhammad Hannan, et al. "Forearm Motion Estimation with Residual Shoulder Motion using Kinematic Synergies and Recurrent Neural Network." The Proceedings of JSME annual Conference on Robotics and Mechatronics (Robomec) 2020. The Japan Society of Mechanical Engineers, 2020." Could you elaborate on the theoretical improvements brought by this study?

Round 2

Reviewer 1 Report

Thanks for addressing all of my concerns, and revising the manuscript accordingly. I have no further comments.